# Familial hypercholesterolaemia and coronary risk factors among patients with angiogram-proven premature coronary artery disease in an Asian cohort

Sukma Azureen Nazli[1,2], Yung-An Chua[1], Noor Alicezah Mohd Kasim[2], Zaliha Ismail[1,2], Ahmad Bakhtiar Md Radzi[2], Khairul Shafiq Ibrahim[2], Sazzli Shahlan Kasim[2], Azhari Rosman[3], Hapizah Nawawi[1,2]*

**1** Laboratory and Forensic Medicine (I-PPerForM), Institute for Pathology, Universiti Teknologi MARA, Selangor, Malaysia, **2** Faculty of Medicine, Universiti Teknologi MARA, Selangor, Malaysia, **3** Institut Jantung Negara (IJN), Kuala Lumpur, Malaysia

* hapizah.nawawi@gmail.com

**Data Availability Statement:** We have uploaded the dataset to a public repository site UK Data

## Abstract

### Background

Familial hypercholesterolaemia (FH) patients have elevated levels of low-density lipoprotein cholesterol, rendering them at high risk of premature coronary artery disease (PCAD). However, the FH prevalence among angiogram-proven PCAD (AP-PCAD) patients and their status of coronary risk factors (CRFs) have not been reported in the Asian population.

### Objectives

This study aimed to (1) determine the prevalence of clinically diagnosed FH among AP-PCAD patients, (2) compare CRFs between AP-PCAD patients with control groups, and (3) identify the independent predictors of PCAD.

### Methods

AP-PCAD patients and FH patients without PCAD were recruited from Cardiology and Specialist Lipid Clinics. Subjects were divided into AP-PCAD with FH (G1), AP-PCAD without FH (G2), FH without PCAD (G3) and normal controls (G4). Medical records were collected from the clinic database and standardised questionnaires. FH was clinically diagnosed using Dutch Lipid Clinic Network Criteria.

### Results

A total of 572 subjects were recruited (males:86.4%; mean±SD age: 55.6±8.5years). The prevalence of Definite, Potential and All FH among AP-PCAD patients were 6%(19/319), 16% (51/319) and 45.5% (145/319) respectively. G1 had higher central obesity, family history of PCAD and family history of hypercholesterolaemia compared to other groups. Among all subjects, diabetes [OR(95% CI): 4.7(2.9,7.7)], hypertension [OR(95% CI): 14.1

Service ReShare (https://reshare.ukdataservice.ac.uk/855685/).

**Funding:** This study was funded by Malaysia Ministry of Higher Education Long Term Research Grant Scheme [RMI/ST/LRGS5/3 (2/2011)], Universiti Teknologi MARA MITRA Grant [600-IRMI/MYRA 5/3/MITRA (003/20170)-1] and Malaysia Ministry of Higher Education Fundamental Research Grant Scheme (FRGS) [600-IRMI/FRGS 5/3 (067/2019)] which were awarded to the corresponding author (HN). The URL of the funders are (http://mygrants.gov.my/main.php?Content=articles&ArticleID=3&IID=, https://rmc.uitm.edu.my/images/Download/Guidelines/G.Dalaman/G.MITRA/GarisPanduanMitra2017.pdf and http://mygrants.gov.my/main.php?Content=articles&ArticleID=1&IID=; respectively). The funders had no role in study design, data collection and analysis, decision to publish, or preparation of the manuscript.

**Competing interests:** The authors have declared that no competing interests exist.

(7.8,25.6)], FH [OR(95% CI): 2.9(1.5,5.5)] and Potential (Definite and Probable) FH [OR(95% CI): 4.5(2.1,9.6)] were independent predictors for PCAD. Among FH patients, family history of PCAD [OR(95% CI): 3.0(1.4,6.3)] and Definite FH [OR(95% CI): 7.1(1.9,27.4)] were independent predictors for PCAD.

## Conclusion

Potential FH is common among AP-PCAD patients and contributes greatly to the AP-PCAD. FH-PCAD subjects have greater proportions of various risk factors compared to other groups. Presence of FH, diabetes, hypertension, obesity and family history of PCAD are independent predictors of PCAD. FH with PCAD is in very-high-risk category, hence, early management of modifiable CRFs in these patients are warranted.

## Introduction

Familial hypercholesterolaemia (FH) is a hereditary disorder of lipoprotein metabolism, predominantly caused by genetic mutation of low-density lipoprotein receptor gene (*LDLR*). Familial hypercholesterolaemia causes elevated low-density lipoprotein cholesterol (LDL-C) and total cholesterol (TC) serum levels resulting in increased risk of coronary artery disease (CAD) [1]. Globally, the prevalence of FH varies between 1:200–1:500 [2] with >50% FH patients are estimated to be residing in Asia Pacific region [3]. In Malaysia, the prevalence of heterozygous FH was reported to be ~1:100 [4,5]. With the Malaysian population size of 33 million (Department of Statistics Malaysia, 2021), it is estimated that 330,000 individuals are affected by FH, majority of whom are underdiagnosed and/or inadequately treated [5].

Dutch Lipid Clinic Network Criteria (DLCC) (World Health Organization, 1999) [6] is the most widely used FH diagnostic criteria, other than Simon Broome's [7], US MEDPED [8] the Japanese FH Management Criteria [9] the AHA Guidelines [10] and the most recent Canada's Make Early Diagnosis Prevent Early Death criteria [11,12]. The DLCC categorised FH patients into three categories (Definite, Probable or Possible FH), according to various criteria.

Cardiovascular disease (CVD), including CAD is the leading cause of mortality and morbidity in Malaysia [13] and globally [14], in both men and women [15]. The latest Malaysian National Health and Morbidity Survey (NHMS) 2019 reported that CAD is the leading cause of death in Malaysia, with hypertension, diabetes and hypercholesterolaemia as the major risk factors (Institute for Public Health, 2020). Coronary angiography is considered to be the gold standard for diagnosing CAD [16]. Upon confirmation with coronary angiography, CAD patients are treated with minimally invasive percutaneous coronary intervention (PCI) stenting procedure or undergo coronary artery bypass graft (CABG) if the coronary artery blockage is extensive.

While it is generally known that the main cause of CAD is atherosclerosis [17], there is no universally accepted for the onset age of PCAD. A study considered the age limit ranging from 35 to 55 years [18] while other studies considered the onset age of PCAD at ≤45 years [19] and <45 years in males and <55 years in females [20]. PCAD used in this study is defined as the occurrence of CAD at the age <55 years in men and <60 years in women [21] as it is part of the diagnostic parameter in DLCC. PCAD can be prevented by controlling modifiable risks factors such as blood pressure, cholesterol levels, smoking, diabetes, obesity, lack of physical activity, diet and stress. However, there are also non-modifiable risk factors such as age, gender, ethnicity, genetic factors and family history of PCAD and hypercholesterolaemia (HC)

[22]. While preventive therapy can be customised to alter particular risk factors, individuals with established CVD are at very high risk of recurrent events. The risk of developing CAD rises with age, but in younger patients, the prevalence of these risk factors tends to differ. Smoking is by far the most frequently associated risk factor in premature CAD [23].

FH is an important cause of PCAD. Given that PCAD prevalence is increasing in Asia, and considering Malaysia has the lowest mean age of onset for PCAD compared to the developed countries [13], early detection, optimal treatment and PCAD prevention are critically important. FH in the absence of atherosclerotic cardiovascular disease (ASCVD) is grouped into high-risk category, whilst ASCVD is categorised as very high-risk category. Therefore, FH patients with PCAD are in extremely high risk and requires immediate attention and aggressive treatment. The increased lifetime exposure to elevated LDL-C amongst FH patients is closely associated with increased PCAD risk, hence, appropriate intensive lipid-lowering therapy and lifestyle modification is recommended.

Being underdiagnosed and undertreated [21], many FH patients are unaware of their condition until being admitted for acute coronary syndrome (ACS) or CAD. In Asia, FH is not routinely screened, even among angiogram-proven PCAD (AP-PCAD) patients hence, the FH data among these patients are scarce. Besides, the prevalence of FH among AP-PCAD patients, status of the other coronary risk factors amongst them in the Asian population have not been well established. Furthermore, the independent predictors of PCAD in FH patients is still unclear. Thus, the objective of this study is to (1) determine the prevalence of clinically diagnosed FH among AP-PCAD patients, (2) compare the coronary risk factors (CRFs) between AP-PCAD patients with FH (G1) and without FH (G2), FH patients without PCAD (G3) and normal controls without FH and PCAD (G4); (3) and to identify the independent predictors of PCAD among all subjects and those with FH. The identification of FH particularly among PCAD patients is vital in elevating patients' awareness, prompt timely intervention and ensures specific preventive measures by recognising the risk factors and manage them accordingly.

## Methodology

### Study design and patient recruitment

This was a comparative cross-sectional study where subjects were recruited from the National Heart Institute (IJN) and Specialist Clinics (Cardiology and Lipid Clinics) and community health screening programmes from the year 2018 to 2019.

The inclusion criteria were male and female Malaysians aged ≥18 years, with AP-PCAD and voluntarily consented to participate in this study. PCAD patients (Age of onset <55 years in males and <60 years in females) were enrolled into this study. Patients were diagnosed as having AP-PCAD based on significant angiogram results, or previous history of CABG and/or PCI procedures. FH was clinically diagnosed using the Dutch Lipid Clinic Network criteria (DLCC). The exclusion criteria were non-Malaysians, pregnancy, and those with secondary hypercholesterolaemia (such as hypothyroidism, nephrotic syndrome, cholelithiasis and chronic renal disease).

Normal control subjects were collected through community health screening programmes. All Malaysians aged ≥18 years were eligible for inclusion into this study, while those with secondary hypercholesterolaemia, pregnant women and non-Malaysians were excluded.

A total of 572 individuals were recruited for the study. Subjects were divided into four groups: G1 (Group 1—PCAD with FH), G2 (Group 2—PCAD without FH), G3 (Group 3—Non-PCAD, and non-CAD, but with FH) and G4 (Group 4—normal controls, without PCAD and CAD, nor FH).

## Definition of terms

Coronary artery disease was defined as those with previous medical history of an abnormal coronary angiogram with stenosis of ≥50% [24] in at least one major epicardial coronary artery, or had underwent PCI, and/or CABG [25].

Hypercholesterolaemia and hypertriglyceridaemia were defined as TC >5.2 mmol/L, and TG >1.7 mmol/L respectively. Low HDL-C was defined as <1.0 mmol/L (males) and <1.2 mmol/L (females). Severely elevated LDL-C level was defined as >4.9 mmol/L, based on high CVD risk categories, according to the 5th edition Malaysia CPG on Management of Dyslipidaemia 2017 [26]. Both type 1 and type 2 diabetes mellitus (DM) were defined as those with fasting or random plasma glucose >7.0 and >11.1 mmol/L respectively for newly diagnosed DM, or those with known or previously diagnosed DM with/ without anti-diabetic medications. Hypertension was defined as systolic blood pressure of ≥140 mmHg and/or diastolic blood pressure of ≥90 mmHg or those with known or previously diagnosed as hypertension with/ without anti-hypertensive medications.

BMI categorisation was classified into underweight, normal, overweight and obese (BMI:<18.5, 18.5–22.9, 23.0–24.9 and ≥25, respectively) [27]. Central obesity was defined as waist circumference (WC) measurement of the ≥90cm in males and ≥80cm in females [28].

Subjects were categorised into Definite, Probable, Possible or Unlikely FH, resulted from DLCC scores of >8, 6–8, 3–5 and 0–2 points, respectively. Patients with DLCC of Definite, Probable and Possible were considered as All FH, while Definite and Probable FH were recognised as Potential FH [2,6]. For those who were on statin and had no baseline LDL-C record, their estimation of untreated LDL-C level was calculated using LDL-c adjustment factor, according to the types and dosage of statin [29]. Patients with baseline LDL-c of <4.0 mmol/L were automatically classified as Unlikely FH.

## Biometric data and biological sample collection

Personal and family medical history, smoking status, body mass index (BMI), waist circumference (WC) and lipid-lowering therapy (types and dosage) were collected from the clinics' database and by on-site measurement. Blood pressure were measured in triplicates after the subjects were seated and rested for 3–5 mins, and the mean of last two readings were regarded as current blood pressure of the subjects [30]. Baseline and current lipid profiles, which include TC, triglycerides (TG), LDL-C and high-density lipoprotein (HDL) were also obtained from the clinics' database. All medical history, biomarkers and biometric data for PCAD subjects (G1 and G2) were specifically selected at the time point before or during the onset of PCAD. Data for the normal control subjects were obtained through standard questionnaire, assisted by trained research assistants and physician on-site. Their blood samples (9 mL) were collected, and serum lipid profiles were analysed.

## Statistical analysis

Data were analysed using IBM SPSS Statistics version 25 (IBM, NY, USA). Continuous data were presented as means (SD) (for parametric test) or median and interquartile range (IQR) (for non-parametric tests), while categorical data were presented as percentages. The significance of differences between the numerical variables was determined by using two-sample t-test and One-Way ANOVA (for parametric tests) or Mann-Whitney and Kruskal Wallis tests (for non-parametric tests). The significance of association between categorical variables was determined by using Chi-squared or Fisher Exact test [31]. Logistic regression was used to describe the association between CRFs and CAD, where variables subjected to univariate

analysis with p value <0.25 were included for multiple logistic regression analysis. All final analyses with p value <0.05 were considered as statistically significant.

### Ethical consideration and written informed consent

Ethical approval was obtained from participating organisations through the respective Institutional Research Ethics Committees [ref: 600-RMI (5/1/6) and IJNEC/03/2012 (6)] prior to commencement of the study. The study was conducted in accordance with the Declaration of Helsinki. All subjects provided written informed consent prior to enrolment into the study.

## Results

### Prevalence of familial hypercholesterolaemia

Out of 319 AP-PCAD patients in this study (G1 and G2), the prevalence of clinically diagnosed Definite FH and Probable FH was 6% (19/319) and 10% (32/319), respectively, which add up into 16% Potential FH. When Possible FH was included on top of Potential FH, All FH was 45.5% (145/319). Almost 30% of the AP-PCAD patients (94/319) were Possible FH and 54.5% (174/319) were Unlikely FH (**Fig 1**).

### Clinical characteristics and risk factors of study population

**Table 1** shows the distribution of participants into groups based on the presence of PCAD and clinical diagnosis of FH according to the DLCC criteria. Malay was the major ethnic across all groups. G1 and G2 (those with PCAD) has higher proportions of obesity (BMI) compared to G3 and G4. The presence of family history of PCAD was also significantly higher in G1 compared to the other groups (p<0.05). Similarly, there was also a significantly high percentage of individuals with family history of HC in G1 compared to G2, G3 and G4 (p<0.05).

TC levels of G1 was significantly higher compared to G2 but significantly lower compared to G3. Pre-treatment LDL-C levels of G1 was significantly higher than G2 and G4, but similar to G3, whom were also FH individuals. Only four patients (3.7%) were presented with TX and

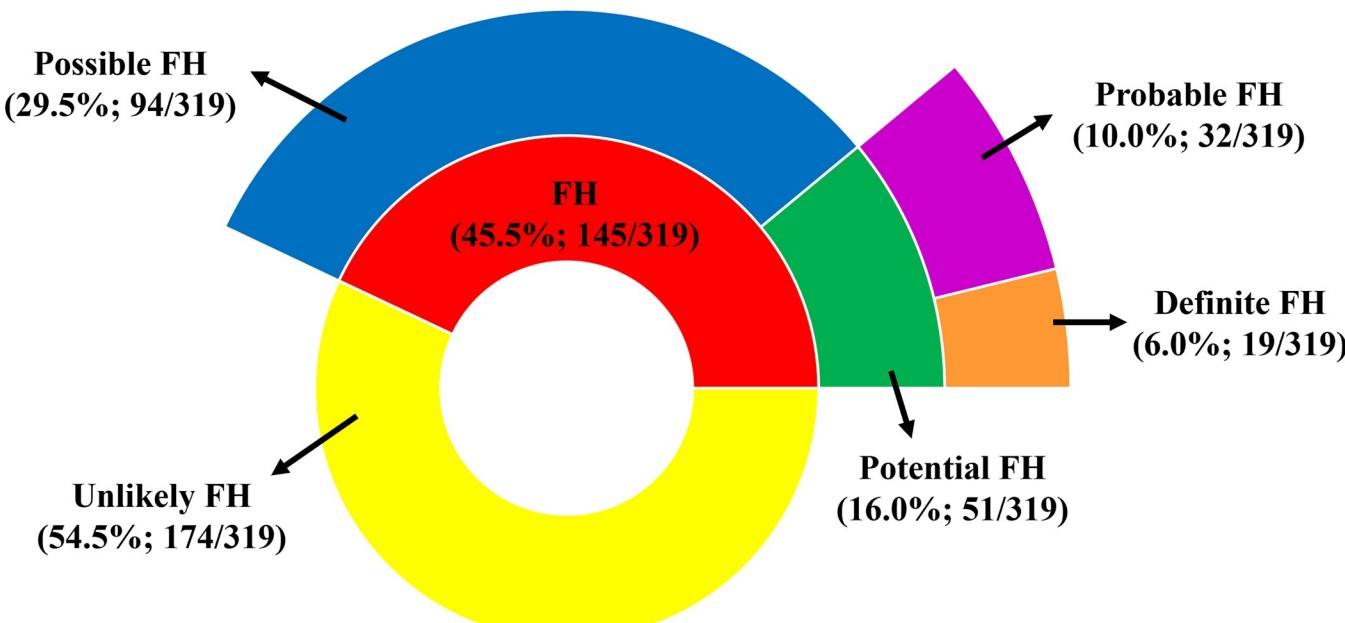

**Fig 1. Prevalence of FH among AP-PCAD individuals (n = 319) based on Dutch Lipid Clinic Network Criteria.**

**Table 1. Distribution of individuals based on the presence of CAD and clinical diagnosis of FH (n = 572).**

| Parameters | G1<br>+FH/+PCAD<br>(n = 145) | G2<br>-FH/+PCAD<br>(n = 174) | G3<br>+FH/-CAD<br>(n = 101) | G4<br>-FH/-CAD<br>(n = 152) | p value |
|---|---|---|---|---|---|
| Gender | | | | | NS^ |
| Male | 128 (88.3) | 144 (82.8) | 89 (88.1) | 133 (87.5) | |
| Female | 17 (11.7) | 30 (17.2) | 12 (11.9) | 19 (12.5) | |
| Age (years) | 54.3 ± 9.3 | 56.8 ± 9.1 | 55.4 ± 8.5 | 55.5 ± 6.8 | NS@ |
| Age range (years) | 26–76 | 35–75 | 42–74 | 46–72 | - |
| Ethnicity | | | | | 0.007^ |
| Malay | 107 (73.8) | 129 (74.1) | 91 (90.1) | 113 (74.3) | |
| Chinese | 15 (10.3) | 26 (14.9) | 2 (2.0) | 15 (9.9) | |
| Indian & Others | 23 (15.9) | 19 (10.9) | 8 (7.9) | 24 (15.8) | |
| BMI# | | | | | <0.001^ |
| Underweight | 0 (0.0) | 0 (0.0) | 3 (3.1) | 1 (0.2) | |
| Normal | 14 (9.9) | 7 (7.1) | 17 (17.3) | 32 (21.5) | |
| Overweight | 15 (10.6) | 19 (19.4) | 12 (12.2) | 31 (20.8) | |
| Obese | 112 (79.4) | 72 (73.5) | 66 (67.4) | 85 (57.0) | |
| Central Obesity# | | | | | 0.037^ |
| Yes | 85 (77.3) | † | 58 (59.2) | 97 (65.1) | |
| No | 25 (22.7) | † | 40 (40.8) | 52 (34.9) | |
| Smoker# | | | | | <0.001^ |
| Current smoker | 40 (27.6) | 28 (16.1) | 25 (26.0) | 44 (29.1) | |
| Ex-smoker | 50 (34.5) | 27 (15.5) | 29 (30.2) | 38 (25.2) | |
| Non-smoker | 55 (37.9) | 119 (68.4) | 42 (43.8) | 69 (45.7) | |
| Diabetes | | | | | <0.001^ |
| Yes | 64 (44.1) | 99 (56.9) | 9 (9.1) | 25 (16.4) | |
| No | 81 (55.9) | 75 (43.1) | 90 (90.9) | 127 (83.6) | |
| Hypertension | | | | | <0.001^ |
| Yes | 93 (64.1) | 151 (86.8) | 28 (28.3) | 39 (25.7) | |
| No | 52 (35.9) | 23 (13.2) | 71 (71.7) | 113 (74.3) | |
| Hypercholesterolaemia | | | | | <0.001^ |
| Yes | 142 (97.9) | 171 (98.3) | 31 (31.3) | 27 (17.8) | |
| No | 3 (2.1) | 3 (1.7) | 68 (68.7) | 125 (82.2) | |
| On lipid lowering therapy# | | | | | <0.001^ |
| Yes | 141 (97.2) | 170 (97.7) | 24 (24.5) | 11 (7.5) | |
| No | 4 (1.8) | 4 (2.3) | 74 (75.5) | 136 (92.5) | |
| Family history of PCAD# | | | | | <0.001^ |
| Yes | 55 (37.9) | 9 (5.2) | 14 (15.4) | 17 (11.7) | |
| No | 90 (62.1) | 165 (94.8) | 77 (84.6) | 128 (88.3) | |
| Family history of HC# | | | | | <0.001^ |
| Yes | 49 (33.8) | 1 (0.6) | 14 (15.2) | 11 (7.6) | |
| No | 96 (66.2) | 173 (99.4) | 78 (84.8) | 134 (92.4) | |
| Tendon xanthomata# | | | | | 0.668^ |
| Yes | 4 (3.7) | 0 (0.0) | 1 (2.0) | 0 (0.0) | |
| No | 105 (96.3) | 2 (100.0) | 50 (98.0) | 34 (100.0) | |
| Corneal arcus (<45 years) # | | | | | <0.001^ |
| Yes | 15 (13.8) | 0 (0.0) | 42 (82.4) | 0 (0.0) | |
| No | 94 (86.2) | 2 (100.0) | 9 (17.6) | 34 (100.0) | |
| *Lipid Profiles* | | | | | |
| TC (mmol/L) | 5.2 (4.5–6.6) [a] | 4.0 (3.5–4.8) [a, b, c] | 6.9 (5.9–7.6) [a, b, d] | 5.5 (4.8–6.2) [c, d] | <0.001* |
| TG (mmol/L) | 1.6 (1.2–2.1) [a] | 1.3 (1.0–1.9) [a, b, c] | 1.8 (1.5–2.5) [a, b] | 1.9 (1.5–2.7) [a, c] | <0.001* |
| LDL-C (mmol/L) | 3.3 (2.6–4.7) [a] | 2.0 (1.6–2.8) [a, b, c] | 5.0 (3.8–5.3) [a, b, d] | 3.4 (2.7–4.0) [c, d] | <0.001* |
| Pre-treatment LDL-C (mmol/L) **, [1] | 5.1 (4.5–5.9) [a, b] | 2.4 (1.7–3.3) [a, c, d] | 5.1 (4.0–5.7) [c, e] | 3.5 (2.6–4.0) [b, d, e] | <0.001* |

(Continued)

**Table 1.** (Continued)

| Parameters | G1<br>+FH/+PCAD<br>(n = 145) | G2<br>-FH/+PCAD<br>(n = 174) | G3<br>+FH/-CAD<br>(n = 101) | G4<br>-FH/-CAD<br>(n = 152) | p value |
|---|---|---|---|---|---|
| HDL (mmol/L) | 1.1 (0.9–1.3) [a] | 1.2 (1.0–1.4) | 1.2 (1.0–1.5) [a, b] | 1.1 (0.9–1.3) [b] | 0.042* |

Data presented as number (n) and percentage (%) for categorical data, mean and standard deviation (SD) and median (interquartile range) [IQR] for continuous data.

^Chi-squared test.

@p<0.05, One-Way ANOVA test.

*p<0.05, Kruskal Wallis test.

**Representing baseline LDL-C level prior to lipid-lowering medication, and LDL-C level for drug-naïve individuals.

#Patients without available data were excluded from the analysis.

[1]Several patients do not have the exact baseline lipid profile–statin conversion was used to calculate the estimated post-treatment LDL-C level.

a, b, c, d, e = p<0.05. Statistical tests with same symbols in a same row are significantly different with each other.

FH: Familial hypercholesterolaemia; PCAD: Premature coronary artery disease; HC: Hypercholesterolaemia; BMI: Body-mass index; LDL-C: Low-density lipoprotein cholesterol; TC: Total cholesterol; TG: Triglyceride; HDL: High-density lipoprotein.

[†]No data available.

15 patients (13.8%) with corneal arcus in G1. Only one individual (2.0%) presented with TX and 42 individuals (82.4%) with corneal arcus in G3 (which were also FH).

## The most common risk factors among individuals with premature coronary artery disease

Individuals with PCAD from G1 (n = 145) and G2 (n = 174) were combined to determine the most common risk factors among PCAD individuals. **Fig 2** shows that after HC (98.1%) and

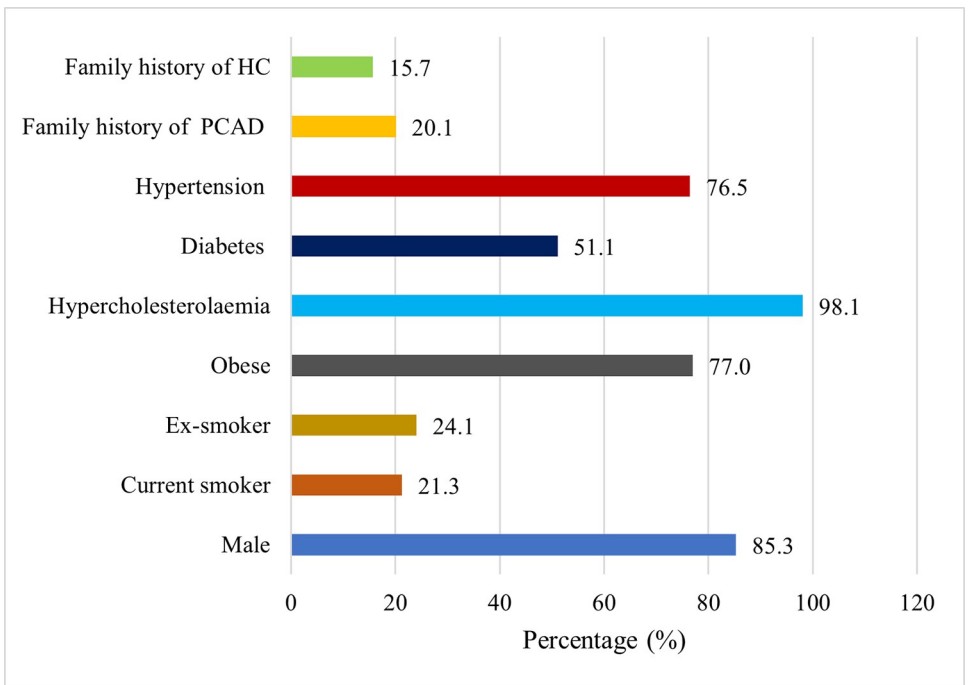

**Fig 2. Conventional risk factors among individuals with PCAD (n = 319).** Data presented as percentage (%); HC: Hypercholesterolaemia, PCAD: Premature coronary artery disease.

Table 2. Factors associated with PCAD among all individuals in the study (n = 572).

| Variables | | Crude OR[a] (95% CI) | Adjusted OR[b] (95% CI) | Wald statistics[b] (df) | p value[b] |
|---|---|---|---|---|---|
| Age | | 1.002(0.982,1.021) | - | - | - |
| Gender (Male) | | 0.740(0.449, 1.221) | - | - | - |
| Ethnicity | Malay | 0.633 (0.421, 0.953) | | | |
| | Chinese | 2.281 (1.231, 4.226) | - | - | - |
| | Indian & others | 1.097 (0.664, 1.810) | - | - | - |
| Smoking | Current smoker | 0.701 (0.476, 1.033) | - | - | - |
| | Ex-smoker | 0.876 (0.598, 1.285) | - | - | - |
| | Non-smoker | 1.438 (1.029, 2.010) | - | - | - |
| Familial Hypercholesterolaemia* | | 1.333 (0.951, 1.870) | 107.034 (16.386, 699.146) | 23.819 (1) | <**0.001** |
| Potential FH | | 3.727 (1.940, 7.158) | 23.164 (3.996, 134.269) | 12.286 (1) | <**0.001** |
| Definite FH | | 5.151 (1.507, 17.611) | | | - |
| Probable FH | | 2.949 (1.380, 6.300) | 0.093 (0.014, 0.622) | 6.001 (1) | **0.014** |
| Possible FH | | 0.825 (0.577, 1.180) | - | - | - |
| Unlikely FH | | 0.750 (0.535, 1.052) | - | - | - |
| Hypercholesterolaemia | | 169.991(71.956, 401.594) | - | - | - |
| Hypertension | | 8.922(6.086, 13.081) | 3.262 (1.366, 7.786) | 7.092 (1) | **0.008** |
| Diabetes mellitus | | 6.776(4.420, 10.388) | 3.928 (1.421, 10.859) | 6.957 (1) | **0.008** |
| BMI Categories | Normal | 0.383 (0.222, 0.662) | - | - | - |
| | Overweight | 0.798 (0.488, 1.305) | - | - | - |
| | Obese | 2.133 (1.435, 3.170) | - | - | - |
| Family history of PCAD | | 1.660(1.041, 2.647) | - | - | - |
| Family history of HC | | 1.576(0.944, 2.632) | 3.093 (1.151, 8.310) | 5.014 (1) | **0.025** |
| High TC[c] | | 0.176 (0.122, 0.254) | - | - | - |
| High TG[c] | | 0.430 (0.306, 0.604) | - | - | - |
| Low HDL-C[c] | | 0.994 (0.679, 1.455) | - | - | - |
| High Pre-treatment LDL-C[c] | | 1.067 (0.724, 1.570) | - | - | - |
| High Post-treatment LDL-C[c] | | 0.447 (0.269, 0.740) | - | - | - |
| On lipid lowering therapy | | 3.556 (2.328, 5.432) | - | - | - |

[a]Simple logistic regression

[b]Multiple logistic regression.

*Familial Hypercholesterolaemia (FH) was defined as individuals that were clinically diagnosed as FH using DLCC criteria. Definite, Probable and Possible were recognised as FH and Unlikely FH were not FH.

OR: Odds ratio; CI: Confidence interval; PCAD: Premature coronary artery disease; BMI: Body mass index; HC: Hypercholesterolaemia; TC: Total cholesterol; TG: Triglyceride; HDL-C: High-density lipoprotein cholesterol; LDL-c: Low-density lipoprotein cholesterol.

[c]High TC: >5.2 mmol/L; high TG: >1.7 mmol/L; low HDL-C: <1.0 mmol/L (males), <1.2 mmol/L (females): High LDL-C: >4.9 mmol/L.

male gender (85.3%), obesity was the most common risk factor among this cohort with (77.0%) followed by hypertension (76.5%) and diabetes (51.1%).

## Association between coronary risk factors and premature coronary artery disease

The factors associated with PCAD among all groups in the study was determined using logistic regression analysis (**Table 2**). All risk factors including lipid profiles were analysed for simple logistic regression. Patients without available data were excluded from the analysis. Further multiple logistic regression analysis shows that only FH, Potential and Probable FH,

**Table 3. Factors associated with PCAD among individuals with clinically diagnosed FH [DLCC score > 3]; (n = 246).**

| Variables | | Crude OR[a] (95% CI) | Adjusted OR[b] (95% CI) | Wald statistics[b] (df) | p value[b] |
|---|---|---|---|---|---|
| Age | | 0.982 (0.954, 1.011) | - | - | - |
| Gender (Male) | | 0.788 (0.336, 1.849) | - | - | - |
| Ethnicity | Malay | 0.190 (0.077, 0.470) | - | - | - |
| | Chinese | - | - | - | - |
| | Indian & others | 2.796 (1.093, 7.152) | - | - | - |
| Smoking | Current smoker | 1.273 (0.667, 2.428) | - | - | - |
| | Ex-smoker | 1.414 (0.763, 2.620) | - | - | - |
| | Non-smoker | 0.742 (0.437, 1.259) | - | - | - |
| Familial Hypercholesterolaemia* | | 97.720 (28.814, 331.415) | - | - | - |
| Potential FH | | 3.753 (1.873, 7.517) | - | - | - |
| Definite FH | | 4.624 (1.329, 16.092) | 9.736 (1.228, 77.214) | 4.640 (1) | **0.031** |
| Probable FH | | 2.706 (1.227, 5.968) | - | - | - |
| Possible FH | | 0.266 (0.133, 0.534) | - | - | - |
| Hypertension | | 4.504 (2.572, 7.888) | 3.310 (1.278, 8.570) | 6.081 (1) | **0.014** |
| Diabetes mellitus | | 8.593 (3.881, 19.025) | 4.644 (1.547, 13.936) | 7.499 (1) | **0.006** |
| BMI Categories | Normal | 0.506 (0.236, 1.083) | | - | - |
| | Overweight | 0.909 (0.398, 2.076) | - | - | - |
| | Obese | 1.871 (1.035, 3.382) | - | - | - |
| Central obesity | | 2.368 (1.293, 4.336) | - | - | - |
| Family history of PCAD | | 3.361 (1.736, 6.509) | 3.303 (1.007, 10.832) | 3.888 (1) | **0.049** |
| Family history of HC | | 2.844 (1.463, 5.529) | - | - | - |
| High TC[c] | | 0.170 (0.087, 0.332) | - | - | - |
| High TG[c] | | 0.749 (0.445, 1.260) | - | - | - |
| Low HDL-C[c] | | 1.809 (0.957, 3.420) | - | - | - |
| High Pre-treatment LDL-C[c] | | 0.751 (0.443, 1.273) | - | - | - |
| High Post-treatment LDL-C[c] | | 0.285 (0.160, 0.507) | - | - | - |
| On lipid lowering therapy | | 4.841 (2.733, 8.573) | - | - | - |

[a]Simple logistic regression

[b]Multiple logistic regression.

*Familial Hypercholesterolaemia (FH) was defined as individuals that were clinically diagnosed as FH using DLCC criteria. Definite, Probable and Possible were recognised as FH and Unlikely FH were not FH.

OR: Odds ratio; CI: Confidence interval; PCAD: Premature coronary artery disease; BMI: Body mass index; HC: Hypercholesterolaemia; TC: Total cholesterol; TG: Triglyceride; HDL-C: High-density lipoprotein cholesterol; LDL-c: Low-density lipoprotein cholesterol.

[c]High TC: >5.2 mmol/L; high TG: >1.7 mmol/L; low HDL-C: <1.0 mmol/L (males), <1.2 mmol/L (females): hHigh LDL-C: >4.9 mmol/L.

hypertension, diabetes and family history of HC were significantly associated with PCAD among all individuals in the study (p<0.05).

Subjects of G1 (n = 145) and G3 (n = 101) was combined (n = 246) to identify the risk factors associated with PCAD among FH individuals using logistic regression (**Table 3**). All risk factors and lipid profiles were analysed using simple logistic regression. Further multiple logistic regression analysis shows that only Definite FH, hypertension, diabetes mellitus and family history of PCAD were significant risk factors for PCAD among those with FH (p<0.05).

Those with Potential FH (Definite: n = 22; Probable: n = 41) from G1 and G3 was combined to identify the risk factors associated with PCAD among Potential FH using logistic regression (**Table 4**). Due to small data, not all risk factors were able to run through regression analysis.

**Table 4. Factors associated with PCAD among individuals with Potential FH [DLCC score > 6]; (n = 63).**

| Variables | | Crude OR[a] (95% CI) | Adjusted OR[b] (95% CI) | Wald statistics[b] (df) | p value[b] |
|---|---|---|---|---|---|
| Age | | 0.896 (0.824, 0.974) | - | - | - |
| Gender (Male) | | 1.257 (0.226, 6.985) | - | - | - |
| Smoking | Current smoker | 3.763 (0.442, 32.042) | - | - | - |
| | Ex-smoker | 0.700 (0.193, 2.535) | - | - | - |
| | Non-smoker | 0.700 (0.198, 2.472) | - | - | - |
| Definite FH | | 1.781 (0.429, 7.403) | - | - | - |
| Probable FH | | 0.561 (0.135, 2.333) | - | - | - |
| Hypertension | | 3.955 (0.957, 16.349) | - | - | - |
| Diabetes Mellitus | | 6.531 (0.781, 54.651) | - | - | - |
| BMI | Overweight | 0.449 (0.037, 5.404) | - | - | - |
| | Obese | 8.800 (2.175, 35.599) | 165.704 (1.360, 20192.347) | 4.349 (1) | **0.037** |
| Central obesity | | 5.911 (1.520, 22.992) | - | - | - |
| Family history of PCAD | | 3.368 (0.893, 12.707) | 14.212 (1.226, 164.704) | 4.508 (1) | **0.034** |
| Family history of HC | | 2.276 (0.550, 9.409) | - | - | - |
| High TG[c] | | 2.080 (0.556, 7.785) | - | - | - |
| Low HDL-C[c] | | 2.727 (0.538, 13.825) | - | - | - |
| High Pre-treatment LDL-C[c] | | 3.200 (0.472, 21.708) | - | - | - |
| High Post-treatment LDL-C[c] | | 0.821 (0.233, 2.893) | - | - | - |
| On lipid lowering therapy | | 1.318 (0.374, 4.647) | - | - | - |

[a]Simple logistic regression

[b]Multiple logistic regression.

OR: Odds ratio; CI: Confidence interval; FH: Familial Hypercholesterolaemia; PCAD: Premature coronary artery disease; BMI: Body mass index; HC: Hypercholesterolaemia; TC: Total cholesterol; TG: Triglyceride; HDL-C: High-density lipoprotein cholesterol; LDL-c: Low-density lipoprotein cholesterol.

[c]High TC: >5.2 mmol/L; high TG: >1.7 mmol/L; low HDL-C: <1.0 mmol/L (males), <1.2 mmol/L (females): High LDL-C: >4.9 mmol/L.

Therefore, only available risk factors were analysed using simple logistic regression. Further multiple logistic regression analysis shows that only obesity and family history of PCAD were significant risk factors for PCAD among Potential FH (p<0.05).

## Independent predictors of premature coronary artery disease

The risk factors that were shown to significantly associated with PCAD were re-analysed using the same regression method. The remaining risk factors were then tested for their interactions and all the factors were re-analysed for the final prediction model of PCAD among all individuals in the study **Table 5(A)**. Overall, hypertension and diabetes were the greatest independent risk factors that contributed to development PCAD [OR (95%CI): 14.1 (7.81, 25.61) and OR (95%CI): 4.7 (2.91, 7.73), respectively], followed by clinically diagnosed as FH and Potential FH [OR (95%CI): 2.9 (1.51, 5.46) and OR (95%CI): 4.5 (2.13, 9.63), respectively].

The final prediction model of PCAD among individuals with FH were shown in **Table 5 (B)**. Results show that FH patients with diabetes have 6.4 times the odds to develop PCAD compared to those without diabetes. Hypertensive patients were 3.6 times more likely to develop PCAD compared to FH patients without hypertension. Besides, those with family history of PCAD have almost 3.0 times the odds to develop PCAD compared to those without family history of PCAD. Those with Definite FH category have 7.1 times the odds to develop PCAD compared to those who were Probable and Possible FH.

**Table 5. Final model of independent predictors of premature coronary artery disease.**

| (A) Independent predictors of PCAD among all individuals (n = 572) | | | | |
|---|---|---|---|---|
| | B(SE) | Wald[a] (df) | OR (95%CI) | *p*-value[*] |
| Diabetes | 1.557 (0.249) | 39.111 (1) | 4.7 (2.91, 7.73) | <0.001 |
| Hypertension | 2.649 (0.303) | 76.383 (1) | 14.1 (7.81, 25.61) | <0.001 |
| FH | 1.054 (0.328) | 10.326 (1) | 2.9 (1.51, 5.46) | 0.001 |
| Potential FH | 1.510 (0.385) | 15.406 (1) | 4.5 (2.13, 9.63) | <0.001 |
| Hosmer and Lemeshow goodness-of-fit test, p-value: 0.914; Percentage prediction: 78.1% | | | | |
| (B) Independent predictors of PCAD among individuals with clinically diagnosed FH [DLCC score > 3] (n = 246) | | | | |
| | B(SE) | Wald[a] (df) | OR (95%CI) | p-value[*] |
| Diabetes | 1.852 (0.431) | 18.465 (1) | 6.4 (2.74, 14.83) | <0.001 |
| Hypertension | 1.293 (0.325) | 15.806 (1) | 3.6 (1.93, 6.89) | <0.001 |
| Family history of PCAD | 1.098 (0.381) | 8.299 (1) | 3.0 (1.42, 6.32) | 0.004 |
| Definite FH | 1.964 (0.687) | 8.178 (1) | 7.1 (1.86, 27.40) | 0.004 |
| Hosmer and Lemeshow goodness-of-fit test, p-value: 0.656; Percentage prediction: 72.9% | | | | |
| (C) Independent predictors of PCAD among individuals with Potential FH [DLCC score > 6] (n = 63) | | | | |
| | B(SE) | Wald[a] (df) | OR (95%CI) | p-value[*] |
| Family history of PCAD | 1.901 (0.880) | 4.668 (1) | 6.7 (1.19, 37.53) | 0.031 |
| Obesity | 2.732 (0.880) | 9.639 (1) | 15.4 (2.74, 86.21) | 0.002 |
| Hosmer and Lemeshow goodness-of-fit test, p-value: 0.150; Percentage prediction: 88.9% | | | | |

[*]Statistically significant at α = 0.05. Model is fit, model assumptions are met, no interaction and multicollinearity problems.

[a] Statistical test: Multiple Logistic Regression.

The final prediction model of PCAD among individuals with Potential FH were shown in **Table 5(C)**. Results show that Potential FH patients with family history of PCAD have 6.7 times the odds to develop PCAD compared to those who were not Potential FH. Besides, patients who were obese were 15.4 times more likely to develop PCAD compared to FH patients who were not obese.

## Discussion

This study is the first to describe the prevalence of clinically diagnosed FH among PCAD cohort in Malaysia, where the prevalence of 16.0% Potential FH is far higher than in the normal population, which had been reported to be about 1% [5]. FH is generally under-diagnosed globally both in the community as well as in hospital settings [21,32], especially in Asian countries. The strength of this present study was that the status of CAD among subjects in G1 and G2 were confirmed by documented angiogram proven PCAD and/or history of angioplasty and/or bypass surgery rather than by self-reporting questionnaire. Besides, physical examinations of TX and corneal arcus were performed by the doctors and physicians during consultation sessions, in contrast to other studies where physical examination of lipid stigmata were not performed for the diagnosis of FH [33].

Currently, the majority of Malaysian FH patients are not genetically confirmed, where genetic testing itself is still not a standard protocol for FH confirmation in the majority of the Asian countries, including Malaysia [34]. Nevertheless, without the molecular testing, the findings show almost half of the AP-PCAD patients were clinically FH.

To the best of our knowledge, this study is also the first report on the prevalence of FH among angiogram-proven PCAD patients in Southeast Asia, although similar studies, also using DLCC criteria had been previously reported in other regions. While the overall

prevalence of FH in this present study is relatively high (45.5%), several studies [32,35] had just reported a similar prevalence of Potential FH (14% - 15%) to this present study where about one third (15%) of AP-PCAD patients were Potential FH and two thirds (30%) were Possible FH.

The prevalence of FH were discovered in lower frequency in several studies, such as a recent study conducted in the United States reported that clinical FH was present in just about 1 out of 10 patients with PCAD (among those with ST-segment elevation myocardial infarction) [36], In contrary, a study in Italy had estimated the prevalence of clinically diagnosed overall FH (including Possible FH) to be very high at 60.4%, with 10.4% of them being Potential FH [37]. Interestingly, the age of onset of PCAD in the latter study was similar to this present study, with patients admitted to cardiac rehabilitation and secondary prevention centres including those with angiogram-proven CAD. Meanwhile, a multi-cohort study in Switzerland with same PCAD age of onset cut-off with this present study, reported the FH prevalence in patients with PCAD as 51.9%, but without definitive angiogram confirmation for PCAD, the proportion of Potential FH is much lower at 4.8% (vs 16.0% in this present study) [38]. This suggests that the detection rate of Potential FH is much higher in those with AP-PCAD or those who had undergone revascularisation procedures, compared to those with ACS without prior angiogram confirmation.

In contrast, a Spanish study reported a much higher prevalence of clinically diagnosed FH among patients with ACS at 77.6%; although with a similar 1:2 ratio between Potential and Possible FH (27.2% of Potential FH and 50.4% Possible FH) [39]. This is possibly due to the higher age cut-off of PCAD used in this study (≤65 years), compared to the present study. Furthermore, this study also determined the prevalence of genetically confirmed FH which was about one-tenth (8.7%) of positive FH rate when clinical diagnosis alone was applied. Our present study did not perform any genetic testing as it not government-funded nor a routine protocol in most Asian countries. Hence, based on the detection rate of the above-mentioned study, the prevalence of genetically confirmed FH among PCAD in this present study could be predicted at approximately 4.6% (one-tenth of 45.5%). However, our study examined for FH in those with AP-PCAD with lower age cut-offs according to DLCN, whilst the Spanish data included all ACS patients with higher age cut-off of ≤65years. Furthermore, the genetic make-up and influence of other modifiable CRFs to development of PCAD may be different between populations. Therefore, future studies are warranted on prevalence of genetically confirmed FH among the AP-PCAD cohort, and the influence of modifiable CRFs to PCAD in these patients in the Asian population.

To date, FH among PCAD patients in hospital settings are still under-diagnosed [40–42]. However, coronary care setting is a useful environment for detecting patients likely to have FH, who should then be referred to specialist service for confirmation of the diagnosis and family cascade screening. Actively diagnosed angiogram-confirmed stenosis is a helpful information in identifying FH among younger PCAD patients, where common physical manifestation of FH such as lipid stigmata is still not identifiable.

Coronary artery disease is a major concern worldwide and CVD is the leading cause of death [43]. Conventional risk factors continue to play a pivotal role in Malaysia. This present study showed that almost 80% PCAD patients with FH (G1) were obese or centrally obese. Previous studies had shown that over 80% of patients with CAD were overweight or obese [44], which was also reflected in this present study, where individuals with AP-PCAD (G1 and G2) had higher combined proportions of overweight and obese individuals when compared to those without CAD (G3 and G4), where the differences were about 10%. This is a concerning finding, where it suggested that FH-CAD group has very suboptimal lifestyle practice that led to obesity, further exacerbating the already-existing high coronary risk within these patients.

On the other hand, G3, despite being diagnosed with FH, had a relatively lower proportion overweight and obesity when compared to G1 and G2, could be explained by the G3 being more health conscious with healthier lifestyle after being diagnosed as FH. Furthermore, this present study clearly showed that obesity is an independent predictor of PCAD in potential FH (p<0.002), suggesting that obesity has an impactful influence on development of PCAD in patients with Potential FH.

In this present study, the level of TC among those with FH-PCAD was significantly lower than G3 (FH without CAD). This could be explained by the fact that most individuals with PCAD were already treated with medications for CAD and statin therapy for at least 3 months prior to the data collection.

There was no positive association between smoking status and type of groups in this present study. The regression analysis of factors associated with CAD also showed that smoking was not associated with PCAD, in contrast with other previous studies [45–47]. However, if ex-smokers are combined with current smokers, the number of this combined categories is significantly higher in subjects in group of interest (G1) compared to G2, G3 and G4 (62.1% vs 31.6% vs 56.2% vs 54.3%: respectively). The high number of ex-smokers in G1 was probably due to the fact that many of those with PCAD ceased smoking upon being diagnosed as PCAD. Ex-smokers may still have residual coronary risk, as reported by a previous study, that 10–15 years of quitting smoking is required before the enhanced smoking-related risk subside, compared to non-smokers [48]. This is another concerning issue in public health as the prevalence of smoking (current and ex-smokers) are very high amongst FH-PCAD patients who are already in the very high-risk category.

Traditional predictors of CAD risk which includes age, gender, smoking, hypertension, diabetes mellitus, family history of CAD, personal history of CAD, HDL-C, TG, small dense LDLs, and lipoprotein(a) [Lp(a)] levels have been studied and reported as potential predictors of atherosclerotic burden and/or CVD prognosis among individuals with FH [49,50].

This present study has demonstrated that among all subjects, diabetes, hypertension, FH and Potential FH were independent predictors for PCAD. In subjects with FH, besides diabetes (OR 6.4) and hypertension (OR 3.6), family history of PCAD (OR 3.0) and having clinically diagnosed as Definite FH (OR 7.1) were the independent predictors for PCAD. This is in agreement with previous reports indicating that family history of CAD in first-degree or second-degree relatives and a personal history of CAD in individuals with FH may cause a higher CVD risk [49]. Besides, those with FH and family history of PCAD were at much higher risk to develop PCAD due to their genetic predisposition to lifelong exposure to elevated LDL-C levels. These findings should alert the clinicians when treating FH patients, who are at even greater CAD risk with the presence of positive family history of PCAD, Definite FH, particularly if they have hypertension and diabetes as co-morbidities. In addition, regression analysis among those with Potential FH revealed that family history of PCAD and obesity were significant predictors of PCAD among this high-risk cohort. It is interesting to note that among those with Potential FH, presence of family history of PCAD and obesity, increase the risk of PCAD by 6.7 and 15.4-fold. Obesity is a well-known risk factor for CVD and is also an important factor in hyperlipidaemias that contributes to lipid phenotypes [51]. In the presence of Potential FH, only obesity and family history of PCAD, but not the other CRFs were clearly shown to be independent predictors of PCAD, suggesting strong synergistic effect of these two CRFs with lifelong exposure to HC in predicting PCAD.

The latest guidelines for the management of dyslipidaemias shows that patients who are clinically diagnosed as FH are automatically categorised into the high-risk category if other risk factors are not present [52]. The risk category is further promoted to very high-risk if ASCVD or one other major risk factor is present. The risk categorisation is important as to

determine the preventive actions based on the patient's total CV risk. The higher the risk, the more intense the medication prescription or treatment and the lower the target LDL-C level. The G1 group in this present study is automatically classified as very high risk due to the inherent presence of PCAD with clinically diagnosed FH. In addition to having FH, they have high proportions of other risk factors such as obesity (BMI≥25), central obesity, diabetes and hypertension. The PCAD risk in these patients are postulated to be further enhanced, in the presence of other risk factors especially hypertension, diabetes, obesity and family history of PCAD. Thus, the treatment and management of FH should be tailored towards those in the very high-risk category, in the presence of ASCVD and/or co-existing CRFs, where management of these co-existing CRFs need to be more intense and optimised. However, the other confounding factors such as the nature of disease-causing genetic variants of FH candidate genes and lipoprotein(a) and their contribution to PCAD have not been included in this present study. Therefore, future studies are warranted to address these issues.

Nevertheless, this study had some limitations where it was not possible to validate all information regarding family history, and physical examination for TX and corneal arcus except those performed in the Specialist Lipid Clinics. Therefore, the prevalence of PCAD patients with FH might be underestimated in this study. This study was also not able to fully describe the prevalence of central obesity among PCAD patients without FH (G2), as the waist circumference data were not available in this cohort. However, the regression analysis of independent predictors for PCAD among FH individuals were not affected. Best efforts had been applied to ensure patients in G3 and G4 were free from CAD during the recruitment. However, because the history of CAD for these groups were collected by means self-reporting questionnaire, patients with asymptomatic cardiac disease such as silent cardiac ischaemia may be wrongly classified, thus affecting the regression analysis. Besides, for these patients (G3/G4), the data of risk factors and other characteristics including the fasting serum lipid results were collected at the entry of study, instead of at the onset of CAD like patients in G1 and G2. For some patients in G1/G2, the history of onset of CAD were collected backdatedly, which explains the age range.

Next, this study population of PCAD patients although recruited from a few major Cardiology and Specialist Lipid Clinics, may not be representative of the national data. This present study also did not include genetic testing for molecular confirmation of FH. High proportion of Possible FH in this study was probably contributed by the active physical examination of corneal arcus. However, only a small portion of these Possible FH subjects were probably genetically true FH, if all of them were genetically tested, as what has been demonstrated in a previous study among clinically diagnosed FH patients in an England lipid clinic, where only 28% of Possible FH patients has genetic mutation in FH-associated genes [53].

## Conclusion

Clinically diagnosed Potential FH is common among AP-PCAD patients in an Asian population. FH-PCAD subjects have high proportions of the various modifiable risk factors. In the FH patients, diabetes, hypertension, obesity, family history of PCAD and Definite FH are independent predictors of PCAD. FH with CAD are in very-high-risk category, hence, early aggressive treatment and management of modifiable CRFs in these patients are warranted for PCAD prevention.

Early identification of these risk factors among FH patients is important in initiating timely intervention. Given that PCAD prevalence is increasing, and more than half of FH worldwide reside in the Asia-Pacific region, early detection, optimal treatment and PCAD prevention are critically needed, especially in Asia. Genetic testing on PCAD patients may also prompt

initiation of family cascade screening, thus promoting early lipid-lowering therapy among affected family members. However, other confounding factors such as the nature of disease-causing genetic variants of FH candidate genes and lipoprotein(a) and their contribution to PCAD have not been included in this present study. Therefore, these issues need to be addressed in future studies. Future research with inclusion of genetic data is warranted to provide more information on the nature of the disease-causing variants of FH-candidate genes, and their association with the other CRFs in predicting PCAD in FH patients.

## Acknowledgments

The authors thank the staffs of IJN and UiTM Specialist Clinics (Cardiology and Specialist Lipid Clinics). We would like to acknowledge Malaysian Health and Well-Being Assessment (MyHEBAT) Study for providing us the normal control samples for this study. We also are indebted to the participants of this study who gave full cooperation and commitment for this study.

## Author Contributions

**Conceptualization:** Hapizah Nawawi.

**Data curation:** Sukma Azureen Nazli, Yung-An Chua.

**Formal analysis:** Sukma Azureen Nazli, Zaliha Ismail.

**Funding acquisition:** Hapizah Nawawi.

**Investigation:** Sukma Azureen Nazli, Noor Alicezah Mohd Kasim, Ahmad Bakhtiar Md Radzi, Khairul Shafiq Ibrahim, Sazzli Shahlan Kasim, Azhari Rosman.

**Methodology:** Sukma Azureen Nazli, Zaliha Ismail, Hapizah Nawawi.

**Project administration:** Noor Alicezah Mohd Kasim, Hapizah Nawawi.

**Resources:** Noor Alicezah Mohd Kasim, Ahmad Bakhtiar Md Radzi, Khairul Shafiq Ibrahim, Sazzli Shahlan Kasim, Azhari Rosman, Hapizah Nawawi.

**Supervision:** Hapizah Nawawi.

**Validation:** Yung-An Chua, Noor Alicezah Mohd Kasim, Hapizah Nawawi.

**Visualization:** Sukma Azureen Nazli.

**Writing – original draft:** Sukma Azureen Nazli, Yung-An Chua.

**Writing – review & editing:** Sukma Azureen Nazli, Yung-An Chua, Noor Alicezah Mohd Kasim, Zaliha Ismail, Ahmad Bakhtiar Md Radzi, Khairul Shafiq Ibrahim, Sazzli Shahlan Kasim, Azhari Rosman, Hapizah Nawawi.

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
