## [Decision Letter · Decision Letter 0]

12 Jan 2022

PONE-D-21-27710Familial hypercholesterolaemia and coronary risk factors among patients with angiogram-proven premature coronary artery disease in an Asian cohortPLOS ONE

Dear Dr. Nawawi,

Thank you for submitting your manuscript to PLOS ONE. After careful consideration, we feel that it has merit but does not fully meet PLOS ONE’s publication criteria as it currently stands. Therefore, we invite you to submit a revised version of the manuscript that addresses the points raised during the review process.

Your manuscript is interesting, but the reviewers have raised several concerns.  Check them carefully and reply clearly and mark clearly what you have changed in the revised version. Please ensure that your decision is justified on PLOS ONE’s publication criteria and not, for example, on novelty or perceived impact.

We look forward to receiving your revised manuscript.

Kind regards,

Katriina Aalto-Setala, Professor

Academic Editor

PLOS ONE

Journal Requirements:

Reviewers' comments:

Reviewer's Responses to Questions

**Comments to the Author**

1. Is the manuscript technically sound, and do the data support the conclusions?

Reviewer #1: Partly

Reviewer #2: Yes

2. Has the statistical analysis been performed appropriately and rigorously? 

Reviewer #1: Yes

Reviewer #2: Yes

3. Have the authors made all data underlying the findings in their manuscript fully available?

Reviewer #1: No

Reviewer #2: Yes

4. Is the manuscript presented in an intelligible fashion and written in standard English?

Reviewer #1: Yes

Reviewer #2: Yes

5. Review Comments to the Author

Reviewer #1: Nawawi et al. conducted a comparative cross-sectional study to examine the prevalence of familial hypercholesterolemia (FH) among angiogram-proven premature CAD (PCAD) patients, the distribution of coronary risk factors, and the prediction of PCAD by the risk factors in Malaysia. The major findings showed that hypertension [OR (95% CI): 14.1 (7.8, 25.6)] and diabetes [OR (95% CI): 4.7 (2.9, 7.7)] carry much more risk than FH [OR (95% CI): 2.9 (1.5, 5.5)] to predict PCAD. However, such an interpretation needs cautions. Other findings add minimal information to current knowledge regarding FH in PCAD, when lacing genetic analysis.

Major critiques:

1. The importance of possible FH is overemphasized. From the analysis of table 5, the odds ratio for potential FH (definite FH + probable FH) is 4.5, and that for FH (definite FH + probable FH + possible FH) is 2.9. This indicates that possible FH is of minimal clinical significance for predicting PCAD. The author should analyze the odds ratio of possible HF, and if it is the case, the part of possible FH should be toned down in the full paper, including the abstract.

2. The definition of premature CAD (PCAD) in this study is age of onset: males: <55; females: <60 years. However, in Table 1, the age range of Group 3 (+FH/-PCAD) is 42 – 74 years and that for Group 4 (–FH/-PCAD) is 46 – 72 years. The age distribution cannot exclude that some male patients younger than 55 years and female patients younger than 60 years may develop PCAD later in their life and should not grouped as -PCAD.

Other comments

1. The definition of PCAD in this study is age of onset: males: <55; females: <60 years. However, in Table 1, the age range of Group 1 (+FH/+PCAD) is 26 – 76 years and that for Group (–FH/+PCAD) is 35 – 75 years. In this case, presence of cardiovascular risk factors should be before the diagnosis of the CAD to be the risk factors. The authors need to clarify the time sequence of appearance of risk factors and onset of PCAD.

2. Similarly, in Figure 2, regarding the percentage of ex-smoker and current smokers, do you count it at the entry of this study, or at the onset of PCAD?

3. In Group 2, 97.7% of patients were given lipid-lowering drugs, however, the difference of LDL-C values between pre-treatment and post-treatment is 0.4 mmol/L, are the data correct? Because 97.2% of patients in Group 1 were on lipid-lowering drugs and the difference of LDL-C values between pre-treatment and post-treatment is 1.8 mmol/L.

Reviewer #2: Review PONE-D-21-27710

Title: Familial Hypercholesterolaemia and coronary risk factors among patients with angiogram-proven premature coronary artery disease in an Asian cohort

Authors: Nazli SA, Chua Y-A, Kasim NAM et al

Background - This paper reports the prevalence of clinically diagnosed FH in Malaysian patients with and without angiographically proven premature coronary artery disease (AP-PCAD) and, in a four-way analysis, assesses the prevalence of coronary risk factors (CRFs) in subjects with and without AP-PCAD by FH status. It concludes by reporting the results of a multiple logistic regression analysis to determine the association between CRFs and CAD in the total study population and separately for those with FH.

Diagnostic criteria - A particular strength of this large study of 572 subjects is its rigorous evaluation of CAD defined as a previous medical history of an abnormal coronary angiogram with stenosis >50% in at least one major epicardial coronary artery or prior PCI and/or CABG in males with an age of onset <55 & females <60 years. FH was diagnosed using the Dutch Lipid Clinic Network criteria (DLCNC), although the authors accept the lack of confirmatory genetic testing for FH-causing mutations is a limitation. Nevertheless, arguably excessive credence is given in the paper to the DLCNC as a basis for a clinical diagnosis of FH. The criteria were initially developed as a means of cost-effectively maximising detection rates for cascade testing programmes by screening only relatives of index patients at high likelihood of genetic FH based on their DLNC score. Identification of an elevated LDL-cholesterol levels in 1st and 2nd-degree relatives would then confirm the likelihood of FH in the index case as well as identifying previously undiagnosed and untreated relatives. As fast-throughput, less expensive, methods of mutation testing became available, DLCN criteria scores have increasingly been used to select patients warranting confirmatory genetic testing. Detection of an FH-causing mutation then initiates cascade testing for that mutation.

Misclassification with DLCN scores - FH mutation detection rates in patients assessed by DLCN scoring has been documented in a number of studies (e.g. Tada H et al. Circulation Journal 2021;85:891-7, & Futema M et al. Atherosclerosis 2013;229:161-8). The former study included cascade-screened patients and those with Achilles tendon thickness measurements, which may have resulted in a positive diagnostic selection bias, whereas the latter smaller study recruited patients sequentially attending a lipid clinic over a three-year period so there may be less risk of bias. The results are summarised below:

Tada H et al Futema M et al

DLCN score n Mutation n Mutation

Diagnosis positive (%) positive (%)

Unlikely <3 367 5 (1) 13 3 (23)

Possible 3-5 156 49 (31) 69 19 (28)

Probable 6-8 57 30 (53) 49 19 (39)

Definite. >8 100 91 (91) 89 48 (54)

Clearly nearly all patients with a score of <3 will not have an FH-causing mutation and over two thirds with a “possible” score of 3-5 will also be unaffected. A mutation will be identified in only about a half with a “probable” score of 6-8, but the majority with a “definite” score of >8 will have an FH-causing mutation. This suggests that the authors’ conclusion in the abstract that “almost half of the AP-PCAD patients with a score of >3 should be classified as “clinically diagnosed FH” is misleading, particularly since “possible FH” (score 3-5) accounted for 65% (94/145) of all patients categorised as having “clinically diagnosed FH”. In fact, their data shows that only 19/319 (6.0%) have a “definite” score >8 indicating a high likelihood of an FH-causing mutation being identified. They might alternatively consider concluding that 16% have “potential” FH (i.e. DLCN score >6) which suggests that about half of such patients may actually have a mutation. However, it is clearly misleading to conclude in the abstract that the prevalence of FH among AP-PCAD patients is 45.5%. This figure should be relegated to the results section of the paper and removed from the abstract. I accept it is appropriate to consider the finding in relation to the existing literature in the discussion, but it is inappropriate to give it such prominence in the abstract.

Results – These are clearly presented both graphically and in tabular form. In Figure 1, to avoid any confusion, it would be helpful to state in the title that the prevalence of FH is based on the DLCN criteria.

The high prevalence (Table 1) of diabetes in patients with PCAD with and without FH defined by a DLCN score >3 (44% & 57% DM respectively) is striking. Diabetes is, of course, well recognised to result in premature cardiovascular disease. A clinical history of premature CAD scores 2 in the DLCN criteria and patients with premature cerebral or peripheral vascular disease score 1. Consequently, any patient with both premature CAD and cerebrovascular disease will score 3 and be classified as “possible FH” regardless of their LDL cholesterol concentration.

Unfortunately, the presentation of the results at present does not allow the reader to assess whether a diagnosis of diabetes is inflating the DLCN estimate of FH. Although type 2 diabetes is usually associated with raised triglyceride levels, low HDL, and with little or no increase in total and LDL cholesterol, nevertheless, given the mean age of the population studied, it would not be surprising to find that many of the PCAD subjects with diabetes had modestly increased pre-treatment LDL concentrations of >4.0 – 4.9 mmol/l, which would result in a DLCN score of 3 for subjects with PCAD even in the absence of any other clinical criteria indicative of FH. Perhaps the lipid profiles for these patients could be added as a supplementary table using the same format as Table 1? It would be particularly helpful to view the triglyceride concentrations since individuals with the lowest TG level (0.4-1.0 mmol/l) were shown to have the highest mutation detection rate (60%) in the study by Marta et al. (op cit) and decreased to 20% in those in the top quartile (2.16-4.3 mmol/l). Consequently, in the Welsh Cascade Screening Programme, a DLCN score is reduced by 2 points for a fasting TG of 2.5-3.5, and more for higher TG concentrations

Tables 2 & 3 depict clearly the factors associated with PCAD in a logistic regression analysis. I would suggest adding to the title of Table 3 with “clinically diagnosed FH with DLCN score >3 and, similarly, for Table 4 add “with a DLCN score >6”. Table 5 very succinctly summarises the findings of the final prediction model. Again, I would be inclined to include the DLNC scores.

Specific points:

Line 60: The prevalence cited of ~1 in 100 is almost high enough to suggest a “founder effect” but the reference is to a secondary source. Could the authors please cite the primary source?

Line 90: It might be helpful to cite a reference for Malaysia having the lowest mean age of onset for PCAD.

Line 117: Inclusion criteria – presumably both type 1 & 2 patients with diabetes were eligible for inclusion? The tables do not provide these data, but it is possible that this information was not available.

Line 125: A total of 572 patients were recruited to the study. Do the authors know how many patients were eligible, the number approached to participate and the participation rate?

Line 129: The inclusion criteria for AP-PCAD are clearly defined, but the criteria for Non-PCAD “controls” do not appear to be specified. Had the “controls” undergone angiography and been shown to have no evidence of stenosis or was this established by questionnaire? It is important to clarify this.

Line 132: Were lipids and lipoproteins measured centrally or by multiple methods locally at hospitals/clinics that referred patients to the National Heart Institute and Specialist Clinics. Were triglyceride measurements fasting? It might be helpful to be rather more specific about the sampling frame from which patients were recruited.

Line 133: Presumably LDL-C concentrations were calculated using the Friedewald formula?

Line 202: The abbreviation CA for corneal arcus is not commonly used. Was it used earlier in the paper?

References – Reference 24 duplicates number 21

Discussion – The discussion is well written and clearly expressed. I wonder, however, whether it would benefit by shortening? I could not find a word count but my impression is the paper probably exceeds 5,000 words after excluding the references. If the discussion was less discursive, it would probably be more impactful – and it deserves to be.

Lines 262-270 start by justifiably focusing on the main findings and the strengths of the study. Nevertheless, I think it is a mistake to assert, as the headline finding, that the prevalence of FH was 45.5% among AP-PCAD patients. Instead, I suggest using as a surrogate for “clinically diagnosed FH” a DLCN score of either >6 [probable + definite FH termed “potential” FH in the paper] or >8 [“definite”- approximating to a genetically confirmed diagnosis]. Including patients with a score of 3-5 [“possible”] introduces diagnostic misclassification and results in over-interpretation of the findings and an unconvincing prevalence estimate of 45.5%. Using this obviously fallible definition undermines the credibility of the study’s findings.

Lines 271-277. Most health care systems do not have access to routine genetic testing for FH. Perhaps one sentence might suffice for this paragraph.

Lines 278-287. This paragraph places the findings in the context of the existing literature and is important. The next paragraph (lines 288-299) suggests that the detection rate of potential FH (score >6) is higher in patients with AP-PCAD, or those who have undergone revascularisation procedures, when compared to those with ACS without prior angiographic confirmation. Perhaps simply citing references here rather than detailing individual studies would suffice?

Lines 300-320. The key fact here seems to be that in this Spanish study (ref 43) the prevalence of genetically confirmed FH in patients with PCAD was not dissimilar to that in the current study using the DLNC score for “definite FH”. A further Northern European Study (44) also reported similar consistent findings.

Lines 321-328. This paragraph highlights the importance of considering the diagnosis of FH in patients with a diagnosis of PCAD in the coronary care setting and the need for referral to specialist clinics for confirmation of the diagnosis and cascade testing. Perhaps it could be somewhat shortened?

The remainder of the discussion concentrates on the interesting results of the logistic regression analyses that examine the independent predictors for PCAD. These findings are consistent with previous studies. Interestingly, obesity was shown to be a significant independent predictor of PCAD in patients with potential FH (DLNC score >6). Northern European registry studies have low rates of obesity in FH patients and cannot, therefore, address this question with adequate statistical power. However, as their populations are becoming increasingly overweight and obese, this is potentially an important insight. Overall, however, the authors may feel this section of the discussion could be shortened.

Before concluding, the authors consider objectively limitations of the study and, perhaps, the diagnostic misclassification associated particularly with lower DLCN FH scores should be added.

Summary – This is an interesting study that extends existing knowledge by assessing the prevalence of clinically diagnosed FH in Malaysian patients with and without premature coronary artery disease. The categorisation of cases was rigorously defined angiographically, but it is not clear whether unaffected patients had undergone angiography with no evidence of stenosis found or whether this was elicited by questionnaire. FH was defined clinically using the Dutch Lipid Clinic Network score which inevitably, in the absence of mutation testing, results in some diagnostic misclassification. This is most marked in the lowest DLCN diagnostic category of “possible FH” with a score of 3-5 in which no FH-causing mutation will be found in over two thirds of patients. It is therefore very misleading to conclude in the abstract that the prevalence of FH among AP-PCAD is 45.5% based on a score of >3 since “possible” FH accounted for 65% of patients with “clinically diagnosed FH”. The multiple logistic regression analysis results are convincing and interesting findings. Overall, the impact of the paper would arguably be strengthened if the discussion was substantially shortened.

6. PLOS authors have the option to publish the peer review history of their article (what does this mean?). If published, this will include your full peer review and any attached files.

Reviewer #1: No

Reviewer #2: **Yes: **Professor H.Andrew W. Neil

---

## [Author Response · Author response to Decision Letter 0]

28 Apr 2022

Thank you very much for the kind and constructive questions and comments. We really appreciate the consideration and attention. Below are our answers to the reviewers:

# We have shortened the group’s name in the manuscript from Group 1 to G1, Group 2 to G2, Group 3 to G3 and Group 4 to G4.

Reviewer 1:

Comment:

Nawawi et al. conducted a comparative cross-sectional study to examine the prevalence of familial hypercholesterolemia (FH) among angiogram-proven premature CAD (PCAD) patients, the distribution of coronary risk factors, and the prediction of PCAD by the risk factors in Malaysia. The major findings showed that hypertension [OR (95%CI): 14.1 (7.8, 25.6)] and diabetes [OR (95% CI): 4.7 (2.9, 7.7)] carry much more risk than FH [OR (95% CI): 2.9 (1.5, 5.5)] to predict PCAD. However, such an interpretation needs cautions. Other findings add minimal information to current knowledge regarding FH in PCAD, when lacing genetic analysis. 

Response:

Some studies had demonstrated that hypertension and diabetes can be a much greater independent risk for coronary artery disease even when compared with lipid biomarkers such as LDL-C level (Chan et al., 2015), which is supposedly the main indicator of FH. 

However, we agree to tone down the finding of hypertension and diabetes by rephrasing the Result section. (Page 22, turquoise highlight)

Reference: Chan DC, Pang J, Hooper AJ, Burnett JR, Bell DA, Bates TR, van Bockxmeer FM, Watts GF. Elevated lipoprotein (a), hypertension and renal insufficiency as predictors of coronary artery disease in patients with genetically confirmed heterozygous familial hypercholesterolemia. International journal of cardiology. 2015 Dec 15;201:633-8.

Comment:

The importance of possible FH is overemphasized. From the analysis of table 5, the odds ratio for potential FH (definite FH + probable FH) is 4.5, and that for FH (definite FH + probable FH + possible FH) is 2.9. This indicates that possible FH is of minimal clinical significance for predicting PCAD. The author should analyze the odds ratio of possible HF, and if it is the case, the part of possible FH should be toned down in the full paper, including the abstract. 

Response:

The independent odds ratio for Possible FH has been calculated in Table 2 and 3, where it turns out that regardless of what is the sample cohort (whether among all subjects or just among FH), the odds ratios for association of Possible FH with PCAD is <1, which means Possible FH has minimal clinical significance for predicting FH. We have toned down the prevalence of All FH by relegate it to the result section only, and further highlight the importance of Potential FH.

Abstract: Page 2-3, turquoise highlight.

Results: Page 10, turquoise highlight. 

Discussion: Page 23-24, turquoise highlight.

Conclusion: Page 29, turquoise highlight. 

Comment:

The definition of premature CAD (PCAD) in this study is age of onset: males: <55; females: <60 years. However, in Table 1, the age range of Group 3 (+FH/-PCAD) is 42 – 74 years and that for Group 4 (–FH/-PCAD) is 46 – 72 years. The age distribution cannot exclude that some male patients younger than 55 years and female patients younger than 60 years may develop PCAD later in their life and should not grouped as -PCAD. 

Response:

The definition of PCAD used in this study was applied to patients in G1 (+FH/+PCAD) and G2 (-FH/+PCAD); in which these patients were recruited cross-sectionally based on retrospective diagnosis of CAD, and more importantly, the CAD is already developed and confirmed by angiography. 

For patients in G3 and G4 (-PCAD), we do not deny that the subjects may develop CAD in younger age (as young as 42 years old in G3, and as young as 46 years in G4). However, we deliberately match the age among G1, G2, G3 and G4 so their ages were not significantly different in order to ensure age-sensitive variables such as blood pressure and corneal arcus will have equal odds across the groups. Nevertheless, we have explained the possibility of recruiting patients with silent asymptomatic ischaemia in G3 & G4 in Discussion section. (Discussion: Page 28, turquoise highlight.)

Comment:

The definition of PCAD in this study is age of onset: males: <55; females: <60 years. However, in Table 1, the age range of Group 1 (+FH/+PCAD) is 26 – 76 years and that for Group (–FH/+PCAD) is 35 – 75 years. In this case, presence of cardiovascular risk factors should be before the diagnosis of the CAD to be the risk factors. The authors need to clarify the time sequence of appearance of risk factors and onset of PCAD. 

Response:

The PCAD onset age is already explained the “Methodology - Study design and patient recruitment” section. Care has been taken in G1 (+FH/+PCAD) and G2 (-FH/+PCAD) where only coronary risk factors that appeared before or during the onset of PCAD were recorded. Additional phrase regarding the sequence of appearance of risk factors has been added in “Methodology - Biometric data and biological sample collection” section (Page 8, turquoise highlight).

Comment:

Similarly, in Figure 2, regarding the percentage of ex-smoker and current smokers, do you count it at the entry of this study, or at the onset of PCAD? 

Response: 

The smoking status were specifically recorded at the onset of PCAD, which should be already sufficiently explained in the correction at “Methodology - Biometric data and biological sample collection” section. (Methodology: Page 8, turquoise highlight.) 

Comment:

In Group 2, 97.7% of patients were given lipid-lowering drugs, however, the difference of LDL-C values between pre-treatment and post-treatment is 0.4 mmol/L, are the data correct? Because 97.2% of patients in Group 1 were on lipid-lowering drugs and the difference of LDL-C values between pre-treatment and post-treatment is 1.8 mmol/L. 

Response:

Yes, the data is indeed correct. The post-treatment LDL-C reduction in G1 seems greater than that in G2 because G1 is consists of FH patients, who has inherently high pre-treatment LDL-C (mean = 5.1 mmol/L) compared to the non-FH G2 (mean = 2.4 mmol/L). According to the Malaysian Clinical Practice Guidelines for Dyslipidaemia, the targeted post-treatment LDL-C for very-high coronary risk patients is 1.8 mmol/L. However, according to our preliminary data (Chua et al., 2021 – abstract in conference proceeding), only <10% of FH patients achieved the targeted LDL-C of 1.8 mmol/L, where large portion of patients only achieved post-treatment LDL-C of 2.6 mmol/L or greater. Non-FH hypercholesterolaemic patients may achieve better post-treatment LDL-C level where 23% of them achieved 1.8 mmol/L (Razman et al., 2021 – abstract in conference proceeding).

This explains why the baseline LDL-C of G1 is higher compared to G2, but the post-treatment LDL-C levels of both groups were similar. 

Reference:

1) Yung-An Chua, Sukma Azureen Nazli, Azhari Rosman, Sazzli Shahlan Kasim, Khairul Shafiq Ibrahim, Ahmad Bakhtiar Md Radzi, Noor Alicezah Mohd Kasim, Hapizah Mohd Nawawi. Prescription pattern and achievement of LDL-C targets with lipid-lowering medications among familial hypercholesterolaemia patients attending Specialist Clinics in Malaysia. The 19th International Symposium on Atherosclerosis, Kyoto, 2021. 

2) Aimi Zafira Razman, Noor Alicezah Mohd Kasim, Zaliha Ismail, Alyaa Al-Khateeb and Hapizah Mohd Nawawi. Sub-optimal prescription of lipid-lowering medications and achievement of LDL-C targets in hypercholesterolaemic individuals in the community in the high and very high cardiovascular risk categories. The 19th International Symposium on Atherosclerosis, Kyoto, 2021.

Reviewer 2

Comment:

Misclassification with DLCN scores - FH mutation detection rates in patients assessed by DLCN scoring has been documented in a number of studies (e.g. Tada H et al. Circulation Journal 2021;85:891-7, & Futema M et al. Atherosclerosis 2013;229:161-8). The former study included cascade-screened patients and those with Achilles tendon thickness measurements, which may have resulted in a positive diagnostic selection bias, whereas the latter smaller study recruited patients sequentially attending a lipid clinic over a three-year period so there may be less risk of bias. The results are summarised below:

Tada H et al Futema M et al

DLCN score n Mutation n Mutation

Diagnosis positive (%) positive (%) 

FH Score Tada H et al Futema M et al

Unlikely < 3 367 5 (1) 13 3 (23)

Possible 3-5 156 49 (31) 69 19 (28)

Probable 6-8 57 30 (53) 49 19 (39)

Definite > 8 100 91 (91) 89 48 (54)

Clearly nearly all patients with a score of <3 will not have an FH-causing mutation and over two thirds with a “possible” score of 3-5 will also be unaffected. A mutation will be identified in only about a half with a “probable” score of 6-8, but the majority with a “definite” score of >8 will have an FH-causing mutation. This suggests that the authors’ conclusion in the abstract that “almost half of the AP-PCAD patients with a score of >3 should be classified as “clinically diagnosed FH” is misleading, particularly since “possible FH” (score 3-5) accounted for 65% (94/145) of all patients categorised as having “clinically diagnosed FH”. In fact, their data shows that only 19/319 (6.0%) have a “definite” score >8 indicating a high likelihood of an FH-causing mutation being identified. They might alternatively consider concluding that 16% have “potential” FH (i.e., DLCN score >6) which suggests that about half of such patients may actually have a mutation. However, it is clearly misleading to conclude in the abstract that the prevalence of FH among AP-PCAD patients is 45.5%. This figure should be relegated to the results section of the paper and removed from the abstract. I accept it is appropriate to consider the finding in relation to the existing literature in the discussion, but it is inappropriate to give it such prominence in the abstract.

Response:

We really appreciate the efforts of Reviewer 2 in thoroughly explained the quoted references for backing up his/her comment. The comment is insightful and fair.

We have reduced the prominence of All FH, highlighted the prevalence of Potential FH, and concluded that Potential FH is clinically more important than All FH in the Abstract and Result sections.

We also revised the Conclusion section, where the prevalence of All FH has been and replaced by mentioning Potential FH is common among Asian population.

However, for comparison purpose, we still mentioned few studies which have similarly high prevalence of All FH in Discussion section. 

Abstract: Page 2-3, turquoise highlight.

Results: Page 10, turquoise highlight. 

Discussion: Page 23-24, turquoise highlight.

Conclusion: Page 29, turquoise highlight. 

Comment:

Results – These are clearly presented both graphically and in tabular form. In Figure 1, to avoid any confusion, it would be helpful to state in the title that the prevalence of FH is based on the DLCN criteria. 

- The comment has been addressed.

Comment:

The high prevalence (Table 1) of diabetes in patients with PCAD with and without FH defined by a DLCN score >3 (44% & 57% DM respectively) is striking. Diabetes is, of course, well recognised to result in premature cardiovascular disease. A clinical history of premature CAD scores 2 in the DLCN criteria and patients with premature cerebral or peripheral vascular disease score 1. Consequently, any patient with both premature CAD and cerebrovascular disease will score 3 and be classified as “possible FH” regardless of their LDL cholesterol concentration. 

Response:

In DLCN, personal premature CAD and personal peripheral vascular disease are grouped into a same group of criteria (Personal Clinical History). According to the rule for counting the DLCN score, only one criterion with the highest point will be counted for the overall score (Nordestgaard et al., 2013). If a patient is with both premature CAD and peripheral vascular, he/she will score 2 points (premature CAD) instead of 3.

In our study design, we have avoided misclassification of FH patients by only including subjects with baseline LDL-C of >4.0 mmol/L into the DLCN scoring. Those with <4.0 mmol/L were automatically classified as Unlikely FH.

We have included the extra information regarding the inclusion criteria in “Methodology – Definition of Terms” section. (Methodology: Page 8, green highlight.)

Reference:

Nordestgaard BG, Chapman MJ, Humphries SE, Ginsberg HN, Masana L, Descamps OS, Wiklund O, Hegele RA, Raal FJ, & Defesche JC. (2013). Familial hypercholesterolaemia is underdiagnosed and undertreated in the general population: guidance for clinicians to prevent coronary heart disease. European Heart Journal 34(45): 3478-3490.

Comment:

Unfortunately, the presentation of the results at present does not allow the reader to assess whether a diagnosis of diabetes is inflating the DLCN estimate of FH. Although type 2 diabetes is usually associated with raised triglyceride levels, low HDL, and with little or no increase in total and LDL cholesterol, nevertheless, given the mean age of the population studied, it would not be surprising to find that many of the PCAD subjects with diabetes had modestly increased pre-treatment LDL concentrations of >4.0 – 4.9 mmol/l, which would result in a DLCN score of 3 for subjects with PCAD even in the absence of any other clinical criteria indicative of FH. Perhaps the lipid profiles for these patients could be added as a supplementary table using the same format as Table 1? It would be particularly helpful to view the triglyceride concentrations since individuals with the lowest TG level (0.4-1.0 mmol/l) were shown to have the highest mutation detection rate (60%) in the study by Marta et al. (op cit) and decreased to 20% in those in the top quartile (2.16-4.3 mmol/l). Consequently, in the Welsh Cascade Screening Programme, a DLCN score is reduced by 2 points for a fasting TG of 2.5-3.5, and more for higher TG concentrations .

Response:

The prevalence of diabetes in Malaysia normal population was 18.3% in normal population (Ministry of Health Malaysia, 2020), while in Japan, it is 12.1% (Fujii et al., 2021), where the difference of prevalence between these two countries is about 33%.

Among FH Japanese patients, the prevalence of FH in Japan to 28.6% (Teramoto et al., 2018), which means the Malaysian prevalence of diabetes among FH at 44.1% calculated (about 35% difference) in this current study is plausible.

References:

1. Ministry of Health Malaysia. (2020). National Health and Morbidity Survey 2019 - Non-communicable Disease: Risk Factors and other Health Problems.

2. Fujii H, Funakoshi S, Maeda T, Satoh A, Kawazoe M, Ishida S, Yoshimura C, Yokota S, Tada K, Takahashi K, Ito K. Eating Speed and Incidence of Diabetes in a Japanese General Population: ISSA-CKD. Journal of Clinical Medicine. 2021 Jan;10(9):1949.

3. Teramoto T, Kai T, Ozaki A, Crawford B, Arai H, & Yamashita S. (2018). Treatment patterns and lipid profile in patients with familial hypercholesterolemia in Japan. Journal of Atherosclerosis and Thrombosis 25(7): 580-592.

Comment:

Tables 2 & 3 depict clearly the factors associated with PCAD in a logistic regression analysis. I would suggest adding to the title of Table 3 with “clinically diagnosed FH with DLCN score >3 and, similarly, for Table 4 add “with a DLCN score >6”. Table 5 very succinctly summarises the findings of the final prediction model. Again, I would be inclined to include the DLNC scores. 

- The comment has been addressed, thank you very much.

Comment:

Line 60: The prevalence cited of ~1 in 100 is almost high enough to suggest a “founder effect” but the reference is to a secondary source. Could the authors please cite the primary source? 

Response:

The comment has been addressed (Introduction: Page 4, turquoise highlight.)

Comment:

Line 90: It might be helpful to cite a reference for Malaysia having the lowest mean age of onset for PCAD. 

Response:

The comment has been addressed. The sentence has been slightly changed. (Introduction: Page 5, turquoise highlight.)

Comment:

Line 117: Inclusion criteria – presumably both type 1 & 2 patients with diabetes were eligible for inclusion? The tables do not provide these data, but it is possible that this information was not available. 

Response:

Yes, both patients with type 1 & 2 diabetes were included in the study. We have added the description in “Methodology – Definition of terms”. (Methodology: Page 7, turquoise highlight.)

Comment:

Line 125: A total of 572 patients were recruited to the study. Do the authors know how many patients were eligible, the number approached to participate and the participation rate?

Response:

For G1 (+FH/+PCAD) and G2 (-FH/+PCAD), the eligibility of patients are as below: 

Eligible patients, n=815

The number approached, n=368

Included (participation rate), n (%) : 319 (86.7)

However, due to the nature of G3 (+FH/-PCAD) and G4 (+FH/-PCAD) which were recruited from convenient sampling among the community, the number of approached subjects were far numerous than what was minimally needed by the statistical analysis. Out of n=4702 non-PCAD subjects, n=253/4702 were age, ethnic and gender-matched and randomly selected for this study. 

Comment:

Line 129: The inclusion criteria for AP-PCAD are clearly defined, but the criteria for non-PCAD “controls” do not appear to be specified. Had the “controls” undergone angiography and been shown to have no evidence of stenosis or was this established by questionnaire? It is important to clarify this. 

Response:

Due to the study design, it was impossible to access the medical records of the controls who solely consisted of subjects recruited from community health screening programmes, where the programmes were held in municipal public halls instead of established health centres. The information on the CAD status among unaffected G3 (+FH/-PCAD) and G4 (-FH/-PCAD) were obtained by means of assisted self-reporting questionnaire without angiography confirmation. 

Comment:

Line 132: Were lipids and lipoproteins measured centrally or by multiple methods locally at hospitals/clinics that referred patients to the National Heart Institute and Specialist Clinics. Were triglyceride measurements fasting? It might be helpful to be rather more specific about the sampling frame from which patients were recruited. 

Response:

The lipids and lipoproteins were measured by multiple methods at each hospitals/clinics (National Heart Institute and Specialist Clinics) - Yes, the triglyceride measures at fasting.

Comment:

Line 133: Presumably LDL-C concentrations were calculated using the Friedewald formula? 

Response:

Yes. Calculated LDL-C derived from total cholesterol, HDL, and triglycerides levels using Friedewald formula is a standard laboratory protocol in Malaysia. We intentionally did not elaborate the details of the lipid profile analysis in order to shorten the Methodology section.

Comment:

Line 202: The abbreviation CA for corneal arcus is not commonly used. Was it used earlier in the paper? 

Response:

The “CA” is not a common abbreviation. We have changed all the “CA” to “corneal arcus” in the manuscript.

Comment:

References – Reference 24 duplicates number 21 

Response:

The comment has been addressed. We have removed the duplicated reference. 

Comment:

Discussion – The discussion is well written and clearly expressed. I wonder, however, whether it would benefit by shortening? I could not find a word count, but my impression is the paper probably exceeds 5,000 words after excluding the references. If the discussion was less discursive, it would probably be more impactful – and it deserves to be. 

Response:

We have shortened the Discussion section slightly, thus making the word count well under 5000 words (From Introduction to Acknowledgement).

Comment: 

Lines 262-270 start by justifiably focusing on the main findings and the strengths of the study. Nevertheless, I think it is a mistake to assert, as the headline finding, that the prevalence of FH was 45.5% among AP-PCAD patients. Instead, I suggest using as a surrogate for “clinically diagnosed FH” a DLCN score of either >6 [probable + definite FH termed “potential” FH in the paper] or >8 [“definite”- approximating to a genetically confirmed diagnosis]. Including patients with a score of 3-5 [“possible”] introduces diagnostic misclassification and results in over-interpretation of the findings and an unconvincing prevalence estimate of 45.5%. Using this obviously fallible definition undermines the credibility of the study’s findings. 

Response:

The prevalence of FH based on All-FH has been de-emphasised in the Abstract, Results and Conclusion section.

Abstract: Page 2-3, turquoise highlight.

Results: Page 10, turquoise highlight. 

Conclusion: Page 29, turquoise highlight. 

Comment:

Lines 271-277. Most health care systems do not have access to routine genetic testing for FH. Perhaps one sentence might suffice for this paragraph. 

Response:

The said lines have been shortened. (Discussion: Page 23).

Comment:

Lines 278-287. This paragraph places the findings in the context of the existing literature and is important. The next paragraph (lines 288-299) suggests that the detection rate of potential FH (score >6) is higher in patients with AP-PCAD, or those who have undergone revascularisation procedures, when compared to those with ACS without prior angiographic confirmation. Perhaps simply citing references here rather than detailing individual studies would suffice? 

Response:

The said lines have been shortened. (Discussion: Page 23, turquoise highlight.)

Comment:

Lines 321-328. This paragraph highlights the importance of considering the diagnosis of FH in patients with a diagnosis of PCAD in the coronary care setting and the need for referral to specialist clinics for confirmation of the diagnosis and cascade testing. Perhaps it could be somewhat shortened? 

Response:

The said lines have been shortened. (Discussion: Page 25, turquoise highlight.)

Comment:

The remainder of the discussion concentrates on the interesting results of the logistic regression analyses that examine the independent predictors for PCAD. These findings are consistent with previous studies. Interestingly, obesity was shown to be a significant independent predictor of PCAD in patients with potential FH (DLNC score >6). Northern European registry studies have low rates of obesity in FH patients and cannot, therefore, address this question with adequate statistical power. However, as their populations are becoming increasingly overweight and obese, this is potentially an important insight. Overall, however, the authors may feel this section of the discussion could be shortened. 

Response:

The entire paragraph starting with “coronary artery disease is a major concern worldwide…” has been slightly shortened.

Comment:

Before concluding, the authors consider objectively limitations of the study and, perhaps, the diagnostic misclassification associated particularly with lower DLCN FH scores should be added.

Response:

Since our FH subjects were all hypercholesterolaemic (baseline LDL-c <4.0 mmol/L), we are quite confident that the subjects with lower DLCN score were not caused by misinterpretation of LDL-C levels.

However, we fully acknowledge that our Possible FH, which is about 2/3 of all FH patients, is relatively high in proportion and many of the Possible FH subjects may not be molecularly true FH if confirm by genetic testing.

The caveat has been mentioned at the end of Discussion section. (Discussion: Page 29, green highlight.)

Comment:

Summary – This is an interesting study that extends existing knowledge by assessing the prevalence of clinically diagnosed FH in Malaysian patients with and without premature coronary artery disease. The categorisation of cases was rigorously defined angiographically, but it is not clear whether unaffected patients had undergone angiography with no evidence of stenosis found or whether this was elicited by questionnaire. 

Response:

As what has been answered for “Line 129” comment, we would like to clarify that the unaffected patients were assumed absent of CAD by means of assisted self-reporting questionnaire, without angiography confirmation. 

Comment:

FH was defined clinically using the Dutch Lipid Clinic Network score which inevitably, in the absence of mutation testing, results in some diagnostic misclassification. This is most marked in the lowest DLCN diagnostic category of “possible FH” with a score of 3-5 in which no FH-causing mutation will be found in over two thirds of patients. It is therefore very misleading to conclude in the abstract that the prevalence of FH among AP-PCAD is 45.5% based on a score of >3 since “possible” FH accounted for 65% of patients with “clinically diagnosed FH”. 

Response:

As what has been previously commented, we have changed the prevalence of FH by emphasising the Potential FH instead of All FH in the Abstract, Results and Conclusion sections.

Abstract: Page 2-3, turquoise highlight.

Results: Page 10, turquoise highlight. 

Conclusion: Page 29, turquoise highlight. 

Comment:

The multiple logistic regression analysis results are convincing and interesting findings. Overall, the impact of the paper would arguably be strengthened if the discussion was substantially shortened.

Response:

We have tried our best to shorten the Discussion section. Thank you very much for all the comments.

---

## [Decision Letter · Decision Letter 1]

25 May 2022

PONE-D-21-27710R1Familial hypercholesterolaemia and coronary risk factors among patients with angiogram-proven premature coronary artery disease in an Asian cohortPLOS ONE

Dear Dr.Nawawi,

Thank you for submitting your manuscript to PLOS ONE. After careful consideration, we feel that it has merit but does not fully meet PLOS ONE’s publication criteria as it currently stands. Therefore, we invite you to submit a revised version of the manuscript that addresses the points raised during the review process.

There are still some specific requests by the reviewers that you did not comment adequately in your reply.  Please check them carefully and reply clearly.

We look forward to receiving your revised manuscript.

Kind regards,

Katriina Aalto-Setala, Professor

Academic Editor

PLOS ONE

Reviewers' comments:

Reviewer's Responses to Questions

**Comments to the Author**

1. If the authors have adequately addressed your comments raised in a previous round of review and you feel that this manuscript is now acceptable for publication, you may indicate that here to bypass the “Comments to the Author” section, enter your conflict of interest statement in the “Confidential to Editor” section, and submit your "Accept" recommendation.

Reviewer #1: (No Response)

Reviewer #2: (No Response)

2. Is the manuscript technically sound, and do the data support the conclusions?

Reviewer #1: Partly

Reviewer #2: Yes

3. Has the statistical analysis been performed appropriately and rigorously? 

Reviewer #1: No

Reviewer #2: Yes

4. Have the authors made all data underlying the findings in their manuscript fully available?

Reviewer #1: No

Reviewer #2: Yes

5. Is the manuscript presented in an intelligible fashion and written in standard English?

Reviewer #1: Yes

Reviewer #2: Yes

6. Review Comments to the Author

Reviewer #1: It has been a long time since the previous comments were sent (September, 2021). This delay in response and the substantial change of the revised manuscript makes this reviewer to read the revised manuscript from the beginning again and find some critical points.

Major critiques

1. This is a very unique cohort that patients with premature CAD account for more than 75% (145+174/145+174+101=76.0%) of all patients with CAD. Is this cohort representative of CAD population in your country?

2. Methodology, how group 4 patients were recruited was not mentioned.

3. Table 1, data of central obesity for group 2 (n = 174) are missing, which is difficult to understand.

4. The patient number in Table 4 is 63, rather small, and the statistical analysis may not be appropriate.

Other comments

Abstract, 2nd sentences, “Subjects were divided into AP-CAD with FH (G1)…” � “Subjects were divided into AP-PCAD with FH (G1)…”

Reviewer #2: The authors have addressed my comments. However, I would very strongly recommend one change to the abstract to include the figures for Definite FH. Lines 34-36 should read "The prevalence of Definite, Potential and All FH among AP-PCAD patients were 6%(19/319), 16% (51/319) and 45.5% (145/319) respectively. It is very important to include upfront the figures for definite patients since these are consistent in findings in non-Asian studies and include patients overwhelmingly likely to have an FH-causing variant.

7. PLOS authors have the option to publish the peer review history of their article (what does this mean?). If published, this will include your full peer review and any attached files.

Reviewer #1: No

Reviewer #2: **Yes: **Professor Andrew Neil

---

## [Author Response · Author response to Decision Letter 1]

29 Jun 2022

Reviewer #1:

1. This is a very unique cohort that patients with premature CAD account for more than 75% (145+174/145+174+101=76.0%) of all patients with CAD. Is this cohort representative of CAD population in your country?

= We would like to clarify that only G1 and G2 are consist of PCAD individuals (145+174= 319). The rests in G3 and G4 (101+152= 253) are those without CAD/PCAD. Therefore, this study does not represent the proportions of PCAD among CAD cases in Malaysia. 

=We have added extra description in the following section to clarify the nature of G3 and G4. 

=Methodology - Study design and patient recruitment: (Page 7, green highlight, Lines 130-132).

2. Methodology, how group 4 patients were recruited was not mentioned.

= Those in group 3 (G3) and 4 (G4) were recruited through community health screening programmes. The methodology was briefly mentioned previously, but now we have added a new paragraph to emphasise the point.

= Methodology - Study design and patient recruitment: (Page 7, yellow highlight, Lines 126-128).

3. Table 1, data of central obesity for group 2 (n = 174) are missing, which is difficult to understand.

= The data of central obesity for G2 (-FH/+PCAD) is missing because we did not take the measurements of waist circumference for those in this group due to the nature of recruitment for this particular group. Although the PCAD patients in G2 were consented for this study, the recruitment began way earlier than G1, G3 and G4, where the measurement of waist circumference was not part of the recruitment protocol. 

=For the overall analysis of regression (Results: Table 2) which involves all groups including G2 (n=572), we have no choice but to exclude central obesity. 

=However, we are still able to include central obesity as one of the analysis variables in regression analysis that involves FH patients in G1 and G3 (Results: Table 3 and Table 4).

=Nevertheless, the absence of central obesity does not adversely affect the regression analysis and the final outcome of this study.

4. The patient number in Table 4 is 63, rather small, and the statistical analysis may not be appropriate.

= Based on the central limit theorem (CLT), the distribution of sample means approximates a normal distribution as the sample size gets larger, regardless of the population's distribution. Sample sizes equal to or greater than 30 are often considered sufficient for the CLT to hold (Mascha et al., 2018). 

=Besides, all data for n=63 individuals were complete and variables from the univariate analysis with p value <0.25 were included in the multivariate analysis. For multivariate analysis, the value of p <0.05 were taken as significant.

=Therefore, we are quite confident that our sample size for Table 4 is sufficient, and its statistical analysis is appropriate.

=Reference:

Mascha, E. J., & Vetter, T. R. (2018). Significance, errors, power, and sample size: the blocking and tackling of statistics. Anesthesia & Analgesia, 126(2), 691-698.

5. Abstract, 2nd sentences, “Subjects were divided into AP-CAD with FH (G1)…” �“Subjects were divided into AP-PCAD with FH (G1)…”

= Thank you very much for the comment. We have corrected the term “AP-CAD” to “AP-PCAD”. 

= Abstract: (Page 2, green highlight, Line 30).

Reviewer #2:

1. The authors have addressed my comments. However, I would very strongly recommend one change to the abstract to include the figures for Definite FH. Lines 34-36 should read "The prevalence of Definite, Potential and All FH among AP-PCAD patients were 6%(19/319), 16% (51/319) and 45.5% (145/319) respectively. It is very important to include upfront the figures for definite patients since these are consistent in findings in non-Asian studies and include patients overwhelmingly likely to have an FH-causing variant.

= Thank you very much for the recommendation. We have added the prevalence for Definite FH as suggested.

= Abstract: (Page 2, turquoise highlight, Lines 34-36).

---

## [Decision Letter · Decision Letter 2]

18 Aug 2022

Familial hypercholesterolaemia and coronary risk factors among patients with angiogram-proven premature coronary artery disease in an Asian cohort

PONE-D-21-27710R2

Dear Dr. NAWAWI,

We’re pleased to inform you that your manuscript has been judged scientifically suitable for publication and will be formally accepted for publication once it meets all outstanding technical requirements.

Kind regards,

Xiao-Feng Yang, MD, PhD, FAHA

Academic Editor

PLOS ONE

Additional Editor Comments (optional):

Reviewers' comments:

Reviewer's Responses to Questions

**Comments to the Author**

1. If the authors have adequately addressed your comments raised in a previous round of review and you feel that this manuscript is now acceptable for publication, you may indicate that here to bypass the “Comments to the Author” section, enter your conflict of interest statement in the “Confidential to Editor” section, and submit your "Accept" recommendation.

Reviewer #1: All comments have been addressed

Reviewer #2: All comments have been addressed

2. Is the manuscript technically sound, and do the data support the conclusions?

Reviewer #1: Partly

Reviewer #2: Yes

3. Has the statistical analysis been performed appropriately and rigorously? 

Reviewer #1: Yes

Reviewer #2: Yes

4. Have the authors made all data underlying the findings in their manuscript fully available?

Reviewer #1: No

Reviewer #2: Yes

5. Is the manuscript presented in an intelligible fashion and written in standard English?

Reviewer #1: Yes

Reviewer #2: Yes

6. Review Comments to the Author

Reviewer #1: This reviewer accepted the responses to my comments from the authors. However, the new information provided in the revised manuscript raises some concerns, as follows:

1. The new information provided in the revised manuscript regarding the recruitment does not include group 3. In the Methodology (lines 119-129), it is clearly described that “The inclusion criteria were male and female Malaysians aged ≥18 years, with AP-PCAD and voluntarily consented to participate in this study…” and that “Normal control subjects were collected through community health screening programmes...” Apparently, groups 1 and 2 are subjects with AP-PCAD, and group 4 is normal control subjects (line 132). However, how subjects of group 3, with FH but without PACD/CAD were recruited is not clear.

2. CAD includes PCAD, but PCAD does not include whole CAD. The new information in the revised manuscript reads that “G3 (Group 3 – Non-PCAD, and non-CAD, but with FH) and G4 (Group 4 - normal controls, without PCAD and CAD, nor FH)…” (lines 131-133). To make it clear to the readers, in Table 1 the “G3 +FH/-PCAD” would be best abbreviated as “G3 +FH/-CAD”. Similarly, in Table 1, “G4 -FH/-PCAD” would be best abbreviated as “G4 -FH/-CAD”.

Reviewer #2: The authors have addressed my comments satisfactorily by including in the abstract the number of subjects with definite familial hypercholesterolaemia

7. PLOS authors have the option to publish the peer review history of their article (what does this mean?). If published, this will include your full peer review and any attached files.

Reviewer #1: No

Reviewer #2: **Yes: **Professor H.Andrew W Neil

---

## [Editor Report · Acceptance letter]

25 Aug 2022

PONE-D-21-27710R2 

Familial hypercholesterolaemia and coronary risk factors among patients with angiogram-proven premature coronary artery disease in an Asian cohort 

Dear Dr. NAWAWI:

I'm pleased to inform you that your manuscript has been deemed suitable for publication in PLOS ONE. Congratulations! Your manuscript is now with our production department. 

Kind regards, 

on behalf of

Dr. Xiao-Feng Yang 

Academic Editor

PLOS ONE